# Bidirectional Interplay Among Non-Coding RNAs, the Microbiome, and the Host During Development and Diseases

**DOI:** 10.3390/genes16020208

**Published:** 2025-02-08

**Authors:** Shanshan Nai, Jiaxian Song, Wenting Su, Xiaoqian Liu

**Affiliations:** State Key Laboratory of Microbial Technology, Shandong University, Qingdao 266237, China; 15701207153@163.com (S.N.); jxsong_2019@163.com (J.S.); 18203399770@163.com (W.S.)

**Keywords:** microbiome, ncRNAs, development, diseases

## Abstract

It is widely known that the dysregulation of non-coding RNAs (ncRNAs) and dysbiosis of the gut microbiome play significant roles in host development and the progression of various diseases. Emerging evidence has highlighted the bidirectional interplay between ncRNAs and the gut microbiome. This article aims to review the current understanding of the molecular mechanisms underlying the crosstalk between ncRNAs, especially microRNA (miRNA), and the gut microbiome in the context of development and diseases, such as colorectal cancer, inflammatory bowel diseases, neurological disorders, obesity, and cardiovascular disease. Ultimately, this review seeks to provide a foundation for exploring the potential roles of ncRNAs and gut microbiome interactions as biomarkers and therapeutic targets for clinical diagnosis and treatment, such as ncRNA mimics, antisense oligonucleotides, and small-molecule compounds, as well as probiotics, prebiotics, and diets.

## 1. Introduction

The human microbiome is a complex and diverse ecosystem composed of microorganisms, including bacteria, archaea, fungi, and viruses, as well as microbial activity components, such as microbial structures, metabolites, mobile genetic elements, and relic DNA [1,2]. There are approximately 100 trillion microbes inhabiting the human body, including the oral cavity, respiratory tract, skin, and, most notably, the gut, where nearly 95% of these microbes reside [3,4]. The gut microbiome participates in various physiological functions, such as nutrient absorption, metabolic regulation, essential vitamin synthesis, as well as intestinal barrier and immune regulation, which affects host functions mainly through nucleic acids, proteins, or metabolites [5,6,7]. However, the homeostasis of the gut microbiome is also shaped by host genetics and epigenetics, as well as environmental factors, such as drugs, diet, and lifestyles [8,9,10,11]. Microbiome dysbiosis not only contributes to gastrointestinal diseases, such as colorectal cancers and inflammatory bowel disease (IBD), but also diseases in distant organs, including neurological disease, cardiovascular disease (CVD), metabolic diseases, and immune diseases, as well as aging [12,13,14,15,16,17].

The human genome encodes a vast array of transcripts that do not give rise to proteins. They are known as non-coding RNAs (ncRNAs) and make up more than 98% of the human genome [18,19,20]. Since the first discovery of ncRNAs in the 1950s [21], a variety of ncRNAs have been reported. The emergence of ncRNAs triggered a revolution, which gained enormous momentum in the early 2000s with the discovery of microRNAs (miRNA) and their many close relatives, highlighting the importance of post-transcriptional events in gene expression, especially in eukaryotes [22]. As more and more ncRNAs are discovered, ncRNAs are being categorized into different types, including miRNA [23], long non-coding RNA (lncRNA) [24], circular RNA (circRNA), transfer RNA (tRNA), and ribosomal RNA (rRNA) [25]. A large number of data have shown that ncRNAs are involved in almost all physiological processes, including embryogenesis, organogenesis, and pathological processes such as cardiovascular disease, neurodegenerative diseases, inflammatory diseases, and metabolic diseases [26,27,28]. ncRNAs have emerged as pivotal regulators of gene expression and are involved in various biological processes; they act as regulators involved in cellular processes such as signal transduction, chromatin remodeling, transcription, and post-transcriptional regulation [25,29].

Recently, ncRNAs were reported to play critical roles in the crosstalk between the gut microbiome and host [23,30,31]. On the one hand, changes in the gut microbiome affect host health by regulating the expression of ncRNAs [20,32,33]. On the other hand, ncRNAs in extracellular vesicles mainly secreted by the host intestinal epithelial cells (IECs) or diet-derived exosomal snRNAs shape gut microbiota composition [34,35,36]. Here, we will comprehensively review current findings regarding the reciprocal interactions between the gut microbiome and ncRNAs, mainly miRNAs, as well as discuss related molecular pathways involved in the gut microbiome–ncRNA axis and how it contributes to host physiology and pathophysiology, including with respect to colorectal cancer, IBD, neurological disease obesity, and CVD. Additionally, we will discuss the potential of microbiome–ncRNA-axis-based therapeutics, offering new insights into various human diseases and how to conduct corresponding interventions.

## 2. The Bidirectional Interplay Between the Microbiome and ncRNAs

In addition to host genetic factors and environmental factors, studies have shown that changes in gut microbiome status can affect host ncRNA expression profiling (Figure 1). Via a comparative analysis of the miRNA expression between germ-free (GF) mice and specific pathogen-free (SPF) mice, nine differentially expressed miRNAs were identified in the ileum and colon [37]. miR-665 was found to be dysregulated during colonization, suppressing the expression of ATP-binding cassette sub-family C member 3 (*Abcc3*) [37]. In a later study, 16 differentially expressed miRNAs targeting genes associated with intestinal barrier and immune regulation were identified in the caeca of GF and conventional male mice [38]. Another study found that compared to other IEC types, the miRNAs in intestinal epithelial stem cells (IESCs) exhibited more distinct expression between GF and conventional states [39]. miR-375 was one of miRNAs that were inhibited by the presence of microbiota, thereby affecting the proliferation of IESCs [39]. Nakata et al. found that in response to commensal bacteria, the expression level of miR-21-5p increased to regulate small GTPase ARF4 and promote the permeability of IECs [40]. In addition, the lncRNA profiles in various tissues from adult GF and SPF mice indicate that the gut microbiota regulates the expression of lncRNAs in the gut and metabolic organs [41]. Differentially expressed lncRNAs targeting genes associated with the cardiac hypertrophy, nuclear factors of activated T cells (*NFAT*), gonadotropin-releasing hormone (*GnRH*), calcium, and cAMP-response element-binding protein (*CREB*) signaling pathways were analyzed in the hippocampus of SPF and GF mice [42]. Adherent-invasive *Escherichia coli* (*E. coli*) colonized the mucosa induced the expression of miR-30c and miR-130a to limit the expression of the autophagy-related genes *ATG5* and *ATG16L1* in Crohn’s disease [43]. A recent, excellent study by Wang et al. demonstrated that gut microbiota repressed the expression of lncRNA Snhg9 (small nucleolar RNA host gene 9) through an immune relay encompassing myeloid cells and group 3 innate lymphoid cells (ILC3), which increased the activity of PPARγ to reprogram intestinal lipid metabolism in epithelial cells of the small intestine [44] (Table 1).

**Table 1 genes-16-00208-t001:** Details regarding how microbiota modulates host ncRNA expression and function.

Microbiota	ncRNAs	Targets	Biological Functions	References
Commensal bacteria	miR-665	*Abcc3*		[37]
Commensal bacteria	miR-351 miR-664 miR-455		regulating barrier-related genes and carrying out procesed involved in immune regulation.	[38]
Commensal bacteria	miR-375		affecting the proliferation of IESCs	[39]
Commensal bacteria	miR-21-5p	ARF4	promoting the permeability of IECs	[40]
Adherent invasive *E. coli*	miR-30c, miR-130a	*ATG5*, *ATG16L1*	regulating autophagy	[43]
Commensal bacteria	lncRNA Snhg9	PPARγ	reprogramming intestinal lipid metabolism	[44]
Probiotic *E. Nissle* 1917	miR-24, miR-200a	SERT	reducing free-serotonin levels	[45]
Commensal bacteria	miR-10a	IL-12/IL-23p40	suppressing IBD T helper (Th)1 and Th17 cell responses	[46,47]
Adherent invasive *E. coli*	let-7b	TLR4	ameliorating the severity of colitis	[48,49]
*L. monocytogenes*	miR-146a			[50]

Accordingly, emerging evidence is showing that fecal ncRNAs produced by the host and secreted into feces can be incorporated by gut microbiota, thereby shaping its composition and functions (Figure 1). Liu et al. found that fecal miRNAs are predominantly produced by IECs and Hopx-positive cells in the gut and enter gut bacteria, thereby affecting bacterial growth and regulating bacterial gene expression [34,35,36]. For example, miR-515-5p elevated the number of 16S rRNA/23S rRNA transcripts in *Fusobacterium nucleatum* (*F. nucleatum*), while the level of *yegH* mRNA in *E.coli* was upregulated by miR-1226-5p [36]. In IEC-miRNA-deficient (Dicer1−/−) mice, the gut microbiota is out of control and colitis worsens, but the former state can be restored through fecal miRNA transplantation [36]. Thus, fecal miRNAs act as a reliable biomarker for several diseases, reflecting fluctuations in intestinal microbes and intestinal pathology [51,52,53,54], which we will discuss in detail in Section 5 (Table 2).

In addition to fecal ncRNA, dietary-derived exosomal ncRNAs can modulate gut microbiota signatures or composition, thereby affecting human or murine physiology (Figure 1) [55,56]. Teng et al. showed that ginger-derived exosome-like nanoparticles contain miRNAs such as mdo-miR7267-3p, which is preferentially taken up by *Lactobacillus rhamnosus* (*L. rhamnosu*) and targets the monooxygenase *ycnE* gene [57]. Another study showed that *Porphyromonas gingivalis* (*P. gingivalis*) selectively absorbs ginger-exosome-like nanoparticles [58]. Colitis in mice can be ameliorated by garlic exosomes rich in peu-miR2916-p3, which promote the growth of the anti-colitic *Bacteroides thetaiotaomicron* (*B. thetaiotaomicron*) [59]. It has been shown that exosome-derived miRNAs in human milk affect the health and development of infants [60,61]. miRNA-148a is dominant in milk exosomes participating in intestinal maturation, barrier function, thermogenesis, and immune regulation [62,63]. Milk-derived exosomal miRNA-30b and miRNA-148a may convert white adipose tissue into beige/brown adipose tissue during thermogenesis by regulating the expression of uncoupling protein 1 [63]. miR-378 and -320 family miRNAs in human milk play roles in adipogenesis that involve in regulating infant growth at 6 months of age [64]. Extracellular vesicles derived from bovine milk can alter the gut microbiota through oral administration, indicating their therapeutic potential in regulating gut microbiota [65,66,67] (Table 2).

**Table 2 genes-16-00208-t002:** ncRNAs that modulate gut microbiota function and composition.

ncRNAs	Microbiota	Targets	Biological Functions	References
miR-515-5p, miR-1226-5p	Commensal bacteria	Bacterial gene transcripts	protecting the integrity of the intestinal epithelial barrier	[36]
mdo-miR7267-3p	*L. rhamnosus*	monooxygenase *ycnE* gene	carrying out processes relating to barrier function improvement	[57]
peu-miR2916-p3	*B. thetaiotaomicron*		regulating intestinal inflammation	[59]
miR-30	*Mycobacterium tuberculosis*	TLR/MyD88 signaling pathway	immune regulation	[60]
miR-146a	Commensal bacteria		immune regulation	[50]
miR-193a-3p	Commensal bacteria	PepT1	ameliorating DSS-induced colonic inflammation	[68]
lncRNA CARINH	Commensal bacteria	IL-18BP	maintaining intestinal homeostasis	[69]

Thus, all these studies illustrate that there is a bidirectional interplay between gut microbiota and ncRNA. However, whether this crosstalk is direct or indirect is still unclear.

## 3. Molecular Pathways Involved in the Microbiome–ncRNA Axis

Although comprehensive studies have shown the bidirectional correlation between the microbiome and ncRNAs, the underlying molecular pathways by which the gut microbiome regulates host ncRNA expression and whereby fecal or dietary-derived exosomal ncRNAs shape the gut microbiome are still unclear. There are two possible mechanisms: metabolic pathways and immune pathways.

### 3.1. Functions of Metabolites in Microbiome–ncRNA Axis

Different metabolites from nutrient substrates are produced by gut microbiota, including short-chain fatty acids (SCFAs), indole derivatives, polyamines, secondary bile acids, crucial vitamins, and tryptophan (Figure 1) [6,70]. Studies are showing that metabolites from gut microbiota may regulate ncRNA functions [20,31,33,71,72,73]. SCFA treatment led to a change in the expression of miR-10a-5p, which binds to the 3′UTR of *pik3ca* and inhibits PI3K-Akt pathway activation, leading to colitis alleviation [74]. Du et al. showed that an increase in butyrate and acetate levels can regulate DNA methylation levels at the miR-378a promoter in mice, thereby preventing obesity and improving glucose intolerance [75]. Administration of the gut-microbiota-derived tryptophan metabolite indole-3-propionic acid (IPA) regulates the miR-142-5p/ABCA1 (ATP-binding cassette transporter A1) signaling pathway, thereby facilitating cholesterol transport and reducing atherosclerotic plaque [76]. It has been reported that polyamines promote Th17 polarization and disease progression via modulating miR-542-5p expression [77]. Li et al. demonstrated that the levels of miR-92a-1-5p were increased in gastric intestinal metaplasia tissues and that its activity was induced by bile acids [78]. Providing recombinant vitamin E to mice reduced their levels of miR-499 and miR-21 and increased the expression of miR-210, thereby alleviating cardiac hypertrophy and fibrosis [79]. In addition to beneficial metabolites from the digestion of nutrients, gut microbiota also produces genotoxic compounds, which may also regulate ncRNA expression. Colibactin is produced by *E*. *coli* in colorectal cancer, inducing the expression of miR-20a-5p, which targets SENP1, a key protein mediating p53 SUMOylation, thereby promoting colon tumor growth [80] (Table 3).

**Table 3 genes-16-00208-t003:** Summary of microbiota-derived crucial metabolites that modulate host ncRNAs.

Metabolites	ncRNAs	Targets	Biological Functions	References
SCFAs	miR-10a-5p	*pik3ca*	inhibitting PI3K-Akt pathway activation	[74]
Acetate and butyrate	miR-378a	YY1	preventing the development of obesity and glucose intolerance	[75]
Indole-3-propionic acid	miR-142-5p	ABCA1	facilitating cholesterol transport	[76]
Polyamines	miR-542-5p	TGF-β/SMAD3	Promoting Th17 polarization	[77]
Bile acids	miR-92a-1-5p	FOXD1	Promoting gastric intestinal metaplasia	[78]
Vitamin E	miR-21, miR-499 miR-210	MAPK, mTOR, PI3K-AKT	alleviateing cardiac hypertrophy and fibrosis	[79]
Colibactin	miR-20a-5p	SENP1	promoting colon tumor growth	[80]

Exosomal ncRNAs derived from fecal transplantation and one’s diet have also been shown to affect the metabolic activity and metabolite production of gut microbiota (Figure 1) [23,57,81,82,83]. mdo-miR7267-3p in ginger-derived exosome-like nanoparticles increased indole-3-carboxaldehyde (I3A) production [57] (Table 2). Treating probiotic *E. coli Nissle 1917* with extracellular vesicles downregulated miR-24 and miR-200a, thereby upregulating SERT expression and reducing free-serotonin levels [45] (Table 1). Oral administration of probiotic-derived nanoparticles increased bile acid production through intestinal miR-194 suppression [78]. Extracellular vesicles derived from *Akkermansia muciniphila* (*A. muciniphila*) are a novel mucosal delivery vehicle for ameliorating obesity, mainly via regulating tight junctions to alter intestinal permeability [84,85]. However, more studies are needed to further reveal how exosomal ncRNAs regulate the gut metabolites of microbiota.

### 3.2. Functions of Immunity in Microbiome–ncRNA Axis

ncRNAs have been reported to play important roles in immune responses via negatively regulating the expression of key immune development genes, thereby affecting innate and adaptive immunity [86,87,88,89]. Some autoimmune diseases and hematological disorders may be caused by abnormal expression of miRNAs [90,91,92]. miR-155 is the first ncRNA identified to have multiple physiological functions, including in regard to inflammatory responses and immune memory [93,94,95]. In addition, it has been reported that miR-182 and miR-146a play important roles in immune regulation in the body [96,97]. miR-181a/b was identified as an intrinsic regulator of T cell and B cell development [98,99,100]. Several studies have demonstrated that ncRNAs are involved in gastrointestinal tract immunity. The proliferation of IESCs and the production of the intestinal mucus layer are regulated by miR-375 [101,102]. Further, recent studies have shown that miR-375 inhibits enteroendocrine lineage development and regulates the proliferation of stem cells in the intestinal epithelium. Biton et al. found that abolishing the induction of miRNAs, including miR-375, through the gut-specific deletion of Dicer1 led to the loss of a key T(H)2 antiparasitic cytokine, RELMβ, in goblet cells, predisposing the host to parasite infection [102,103]. IL-23 participates in the inflammatory response of intestinal Th17 cells under the negative feedback regulation of miR-222 and miR-221 [104]. CircHIPK3 promotes intestinal epithelium repair by inhibiting the function of miR-29b, thus increasing the expression of *Rac1*, *Cdc42*, and *cyclin B1* after a wound is generated [105]. LncRNA uc.173 promotes renewal of the intestinal mucosa and epithelial barrier by inducing the degradation of miR-195 and miR-29b, respectively [106,107] (Figure 2).

In addition to ncRNAs, the gut microbiota has also been reported to play critical roles in regulating gut immune functions by maintaining a barrier to increase permeability selectivity for commensal microbes and their metabolites [108,109,110,111,112]. For example, Mortha et al. revealed that the commensal microbes promote crosstalk between RORγt (+) ILCs and macrophages, leading to immune homeostasis in the intestine [113,114]. ILC3s induce immunological tolerance related to microorganisms and intestinal health by selecting antigen-specific RORγt+ Treg cells and Th17 cells [115,116]. The metabolites produced by gut microbiota, such as SCFAs, have been reported to affect the polarization and activation of helper T cells as well as the production of cytokines and neutrophil chemoattractants, including TNFα, IL17, CXCL1, and CXCL8 [117,118,119,120]. Several bile-acid-derived metabolites of gut microbiota have recently been found to play a role in intestinal immunity [121,122]. The secondary bile acid 3 β-hydroxydeoxycholic acid (isoDCA) acts on dendritic cells, enhancing the induction of Foxp3, thereby reducing its immune-stimulatory properties and increasing the number of colonic RORγt-expressing Treg cells [123]. Lipopolysaccharides (LPSs), the bacterial surface glycolipids produced by Gram-negative bacteria in the gut, bind to toll-like receptor (TLR)-4 to activate innate immune responses and inflammation in the gut [109,124,125].

Although it is well known that both ncRNAs and gut microbiota play critical roles in regulating gut immune function, there are limited studies showing directed interplay between the gut microbiota–ncRNA axis and gastrointestinal tract immunity [31,126,127]. Gut microbiota downregulate the expression of miR-10a, which targets IL-12/IL-23p40 in dendritic cells and suppresses IBD T helper (Th)1 and Th17 cell responses [46,47] (Table 1). Studies have reported that extracellular vesicles from milk contain a certain number of immune-related microRNAs, which enhance intestinal immunity and remodel the gut microbiota [66,128,129,130]. In addition, several studies have also revealed crosstalk between the gut microbiota–ncRNA axis and immunity in response to bacterial and viral infections [127,131,132]. miRNA let-7b was found to ameliorate the severity of colitis via modulating Toll-Like Receptor 4 (*TLR4*) expression in IECs in response to adherent-invasive *E*. *coli* infection [48,49] (Table 1). In response to *Listeria monocytogenes* (*L. monocytogenes*) infection, several miRNAs, including miR-194, miR-143, miR-148a, miR-200b, miR-378, and miR-146a, were differentially expressed in the gut in GF mice and conventional mice [133]. miR-146a helps maintain gut microbiota homeostasis after *L*. *monocytogenes* infection [50] (Table 1). Antibiotic interference alters gut microbiota and affects the expression of miR-146b and miR-29c in the lungs, ultimately hindering host antiviral immunity [134]. Recent studies have shown that miRNAs mediate the interaction between COVID-19 and host-microbiota homeostasis, thereby reflecting disease severity [135,136,137,138].

These studies have demonstrated that both metabolites and immunity are involved in the crosstalk between gut microbiota and ncRNAs. Of note, none of the pathways are independent, and they are closely related to each other, as discussed above with respect to the metabolites derived from microbiota regulating immune responses, whereas immunity alters the production of metabolites via shaping microbiota composition and intestinal homeostasis. Therefore, it is worth further investigating the molecular pathways through which these processes occur in greater detail.

## 4. The Roles of Microbiome–ncRNA Axis During Development

Numerous studies have illustrated the crucial functions of ncRNAs, as well as other epigenetic mechanisms, including chromatin accessibility, DNA methylation, and histone modification, in both developmental and differentiation processes, such as stem cell maintenance, embryogenesis, organogenesis, dosage compensation, and genomic imprinting [139,140,141,142,143,144,145,146]. Zebrafish mutants lacking maternal and zygotic Dicer activity undergo gametogenesis, cell fate determination, and early patterning, yet they are defective in germ layer formation, morphogenesis, and organogenesis. Importantly, mature miR-430 ameliorates many of these defects, suggesting that the phenotype of the MZdicer fish is indeed caused by the loss of miRNA function [147,148,149]. The development of oocytes in Dgcr8 mutant mice is normal, [150], but there are a large number of dysregulated transcripts in Dicer1 mutant oocytes [151,152]. LncRNAs such as RUS have been shown to drive gene expression programs towards neurogenesis, highlighting their importance in neural lineage commitment and brain development [153]. Additionally, lncRNAs like TUNA control pluripotency and neural lineage commitment, indicating their evolutionary conservation and functional significance [154]. LncRNA HoxBlinc activates *Hox* gene expression patterns and mesoderm lineage development through the recruitment of Set1/MLL complexes [155]. It is also well known that the gut microbiota is essential during early stages of life, as it modulates the host’s immune system as well as influences host development and physiology, including organ development and morphogenesis and host metabolism [11,156,157]. The composition of gut microbiota throughout a human’s life undergoes three stages: birth to weaning, weaning to the start of a normal diet, and old age. The human gut microbiome is formed during birth, and it is influenced by various factors, such as the mode of birth (caesarean or natural), gestational age at birth, infant feeding method (breast milk or formula), weaning, and maternal microbiota [158,159,160]. As mentioned above, the extracellular vesicles from breast milk and early-life nutrients contain a number of microRNAs, which shape the infant microbiota and function in the regulation of development [66,128,129,161,162].

## 5. The Crosstalk Between Microbiome and ncRNA in Diseases

The unique expression pattern of ncRNAs in diseases makes them reliable indicators of underlying lesions and provides a wealth of potential biomarkers for diagnosis and prognosis [163]. Below, we will summarize the publications about the roles of the microbiome–ncRNA axis in several microbiome-associated diseases (Figure 3, Table 4).

### 5.1. Colorectal Cancers

As studies have shown that specific ncRNAs serve as biomarkers and potential therapeutic targets of cancers, emerging evidence is showing that ncRNAs play important roles in crosstalk between microbiomes and carcinogenesis. To investigate the interaction between microbiota and miRNAs in a tumor environment, researchers have compared colorectal cancer samples with samples of adjacent cancerous bowel tissue and identified 76 differentially expressed miRNA, including miR-182, miR-503, and miR-17~92, which are the known oncogenic miRNAs [20,164]. Data from the competitive endogenous RNA (ceRNA) network show that lncRNAs form the dominant part of the probiotic or pathogen-mediated ceRNA network, along with some miRNAs and mRNAs, all of which are closely related to microbially mediated cancer occurrence and development [97]. In addition, several studies also showed that the microbiome modulates the expression of ncRNA during colorectal carcinogenesis. *F. nucleatum* inhibited the expression of miR-18a and miR-4802, which are involved in autophagy-related pathways that may contribute to chemotherapy resistance in individuals with colorectal cancer [165]. Moreover, another study demonstrated that *F. nucleatum* can promote the metastasis of colorectal cancer [166]. This process occurs by means of the miR–1322/CCL20 axis and the polarization of M2-type macrophages. *Parvimonas micra* (P. *micra*) also promotes cancer development by upregulating miR-218-5p expression to activate the Ras/ERK/c-Fos pathway [167]. The metabolites of intestinal microflora, including butyrate, can modulate the expression of host ncRNAs such as miR-92a and miR-192-5p to influence the progression of colorectal cancer [168,169]. It has been reported that exosomal miR-200c-3p negatively regulates the migration and invasion of LPS-stimulated colorectal cancers [170] (Figure 3, Table 4).

### 5.2. Inflammatory Bowel Disease (IBD)

The combined incidence of IBD is approaching 0.5% in developed countries and more than 6 million cases per year worldwide [171]. Accordingly, it is expected to become a major global health and policy issue. The etiology of IBD is known to be complex, influenced by genetic factors and involving dysregulated interactions between the gut immune system and the microbiome [172]. It has been shown that differential expression of miRNAs in the feces of IBD patients affects the progression of IBD by regulating the growth of a number of bacteria [173]. miRNAs such as miR-199a, miR-1226, miR-548a, and miR-515-5p affect the pathogens *F. nucleatum* and *E. coli*, as well as the probiotic bacterium *segmental filamentous bacteria* (*SFB*), ultimately leading to the development of IBD [173]. The functions of miRNAs in the gut are diverse. miR-21 can exacerbate dextrose sodium sulphate (DSS)-induced colitis by affecting the gut microbiota [174], while miR-193a-3p has been reported to ameliorate DSS-induced colonic inflammation by affecting the absorption of bacterial products via targeting PepT1 [68,175]. The C15ORF48/miR-147-NDUFA4 axis has been shown to bridge mitochondrial metabolism and inflammation, making it essential in intestinal homeostasis [176]. Accordingly, antibiotic treatment abrogates the effects of miR-193a-3p in DSS-induced enterocolitis [177]. In addition, *L. fermentum* and *L. salivarius* can increase the expression of various miRNAs (including miR-223, miR-155, miR-150, and miR-143) that improve intestinal barrier and microbial homeostasis, thereby alleviating DSS-induced colitis in mice [178]. As it is known that dietary habits determine the core microbiota, the function of dietary miRNAs can also shape the gut microbiota. Ginger exosome nanoparticles (GELNs) containing miRNAs were reported to alleviate DSS-induced colitis in mice by altering their intestinal microbiota [57]. miR-7267-3p in the GELNs promoted the growth of *Lactobacillus* family bacteria and *Bacillus-like bacteria S24-7* and inhibited the growth of *Clostridium difficile* (*C. difficile*), resulting in an improvement in intestinal barrier function [57]. The role of lncRNAs in IBD has been less studied than that of miRNAs. A study identified an IBD-related lncRNA, CARINH, which plays roles in maintaining intestinal homeostasis by interacting with the intestinal microbiota and upregulating the expression of IL-18BP (Figure 3, Table 4) [69]. These results suggest that interactions between gut microbiota and ncRNAs are important for regulating intestinal homeostasis. Emphasizing these communications is essential for the treatment of intestinal diseases. Future studies should focus on accelerating the prediction and validation of ncRNA targets to fully harness the potential of candidate ncRNAs in the context of IBD.

### 5.3. Neurological Diseases

The gut microbiome is altered in patients with neurological diseases, including multiple sclerosis (MS), Alzheimer’s disease, Parkinson’s disease, and autism [179,180,181]. Several studies have revealed the roles of ncRNAs in the microbiota–gut–brain axis. The transfer of feces collected at the peak of disease from an experimental autoimmune encephalomyelitis (EAE) model of MS revealed that miR-30d was enriched in feces from patients with peak EAE and untreated MS [182]. Oral administration of synthetic miR-30d ameliorated EAE by amplifying regulatory T cells (Tregs), suggesting that feces from diseased individuals may be enriched in miRNAs with therapeutic properties [182]. A study showed that lower microbial diversity and numbers of *Faecalibacterium*, unidentified *Ruminococcaceae*, and *Alistipes*, as well as a greater abundance of *Proteobacteria* and *Gammaproteobacteria*, were found in participants with mild cognitive impairment compared with healthy controls, a finding that was correlated with levels of ncRNAs such as hsa-let-7g-5p, hsa-miR-107, and hsa-miR-186-3p [183]. In addition, LPSs produced by microbiota up-regulate miRNAs controled by the pro-inflammatory transcription factor complex NF-kB, including miRNA-30b, miRNA-34a, miRNA-146a, and miRNA-155, in Alzheimer’s disease [184,185,186]. The administration of *L. plantarum PS128 (PS128)* significantly improved motor deficits in PD-like mice, restored altered fecal microbiota composition, and reduced the levels of miR-155-5p, which targets cytokine signaling 1 (SOCS1) to inhibit neuroinflammatory responses (Figure 3, Table 4) [187]. Yu et al. have identified a total of 270 genera and 798 miRNAs with high correlations in the fecal samples of patients with major depressive disorder, providing clues for the interaction between ncRNAs and gut microbiota via the microbiota–gut–brain axis [188]. The pathways through which signals are transmitted within the microbiota–gut–brain axis include the immune pathway, the neuronendocrine pathway, and the autonomic nervous system pathway [189]. Bioinformatic analyses have shown that these pathways are also regulated by fecal miRNAs, which are highly correlated with specific species [188,190]. Communication between ncRNAs, the microbiome, and the host constitutes a complex regulatory network. Therefore, further investigations are necessary to elucidate the molecular mechanisms of ncRNAs in the microbiota–gut–brain axis.

### 5.4. Obesity

An altered human gut microbiome is one of the risk factors for obesity. Researchers have found an increased abundance of bacteria of the class *Bacilli* and its families *Lactobacillaceae* and *Streptococcaceae* and a decreased abundance of several groups within the class *Clostridia*, including *Dehalobacteriaceae, Clostridiaceae,* and Christensenellaceae, in participants with obesity [191]. Using next-generation sequencing, it has also been shown that obese individuals have significantly higher bacterial diversity compared to non-obese individuals, and the composition of the gut microbiota is also changed in obese individuals [192]. A total of 12 gut bacterial species and 26 circulating miRNAs were found to be differentially expressed between obesity cases and controls, among which three miRNAs were correlated with *Bacteroides eggerthi* (*B. eggerthi*) and BMI levels [193]. Metabolites from mouse gut microbiota can regulate energy consumption and insulin susceptibility, mainly by controlling the expression of miR-181 family members in white adipocytes through tryptophan metabolites [194]. Dysregulation of the gut microbiota–miR-181 axis was also found to be necessary for the development of white adipose tissue (WAT) inflammation, insulin resistance, and obesity in mice [194]. In addition, dysregulation of plasma tryptophan metabolite levels in obese children with WAT is associated with miR-181 expression, indicating that the miR-181 family is an important regulatory factor in WAT function [194]. Mishra et al. found that the microbiota of both obese mice and humans showed a reduced capacity to metabolize ethanolamine, resulting in ethanolamine accumulation in the gut [195]. Elevated ethanolamine levels enhance the binding of ARID3a to the miR-101a-3p promoter and increase its expression, leading to a decrease in the stability of *zona occludens-1* mRNA, which in turn weakens intestinal barriers and induces gut permeability and inflammation [195]. It has also been reported that cherry juice restored production (with respect to a decrease) of SCFAs by regulating the composition and abundance of the gut microbiota and suppressed the expression of obesity-associated miRNAs, such as miR-223-3p, miR-200c-3p, miR-132-3p, and miR-125a-5p in high-fat-diet-induced obese mice (Figure 3, Table 4) [196]. All these studies suggest ncRNAs can be used as a therapeutic target to treat obesity.

### 5.5. Cardiovascular Disease (CVD)

Globally, CVD is the leading cause of death, and lipid abnormalities, high blood pressure, and diabetes mellitus are significant causes of CVD [197]. There is a very close relationship between the heart and the intestines, and it is manifested in the crucial role of metabolites produced by gut microbiota in the pathophysiology of heart disease. The results of a metagenomic association study involving stool samples from patients with atherosclerotic cardiovascular disease (ACVD) showed that the abundance of *Streptococcus spp.* and *Enterobacteriaceae* in the intestines of ACVD patients was higher and the metabolism or transportation of various compounds related to cardiovascular health was weakened [198]. It is worth noting that case–control studies on patients with coronary artery disease (CAD) have shown that the gut microbiome may be a contributing factor for CAD [199]. Probiotics and/or fecal microbiota transplantation may facilitate changes in the gut microbiota by altering the production of ncRNAs and SCFAs, thereby enhancing cardiovascular health [200]. It has been reported that trimethylamine n-oxide (TMAO) modulated the expression of miR-17/92, which promoted inflammation and atherosclerosis and CVD, via regulating IL-12A and blood clotting [201]. Protocatechuic acid produced by gut microbiota by metabolizing Cyanidin-3-O-β-glucoside (Cy-3-G) promoted reverse cholesterol transport and regressed atherosclerotic lesions in mice via repressing miRNA-10b [202]. Vikram et al. revealed that the microbiome regulated vascular microRNA-204 levels, which impaired endothelium-dependent vasorelaxation via downregulating Sirtuin1 (Figure 3, Table 4) [203]. More studies are needed to investigate the roles of the ncRNA–microbiome axis in CVD.

**Table 4 genes-16-00208-t004:** Summary of ncRNAs and microbiota or derived metabolites in diseases.

Disease	Microbiota	ncRNAs	Targets	Biological Functions	References
Colorectal cancers	*Firmicutes Bacteroidetes Proteobacteria*	miR-182 miR-503 miR-17~92			[20,164]
		lncRNA LINC00355 lncRNA KCNQ1OT1 lncRNA LINC00491 lncRNA HOTAIR		associating with poor survival	[97]
	*F. nucleatum*	miR-18a miR-4802		taking part in autophagy-related pathways that may contribute to chemotherapy resistance in colorectal cancer	[165]
	*F. nucleatum*	miR-1322	CCL20	promoting colorectal cancer metastasis	[166]
	*P. micra*	miR-218-5p		activating the Ras/ERK/c-Fos pathway	[167]
	Butyrate of intestinal microflora	miR-92a miR-192-5p		influencing the progression of colorectal cancer	[168,169]
		miR-200c-3p		negatively regulating the migration and invasion of lipopolysaccharide-stimulated colorectal cancer	[170]
Inflammatory bowel diseases	*F. nucleatum*, *E. coli*, *SFB*	miR-199a miR-1226 miR-548a miR-515-5p		leading to the development of IBD	[173]
		miR-21		exacerbating DSS-induced colitis by affecting the gut microbiota	[174]
		miR-193a-3p	PepT1	affecting the absorption of bacterial products	[68,175]
		C15ORF48/miR-147	NDUFA4	acting as an indispensable regulator of gut homeostasis, bridging mitochondrial metabolism and inflammation	[176]
	*L. fermentum*, *L. salivarius*	miR-155 miR-223 miR-150 miR-143		improving intestinal barrier function and the balance of intestinal microbiota	[178]
	*Lactobacillus*, *Bacillus-like bacteria S24-7*, *C. difficile*	miR-7267-3p		improving intestinal barrier function	[57]
		lncRNA CARINH	IL-18BP	playing roles in maintaining intestinal homeostasis	[69]
Neurological diseases		miR-30d	T cells	ameliorate EAE	[182]
	*Faecalibacterium*, unidentified *Ruminococcaceae*, *Alistipes*, *Proteobacteria, Gammaproteobacteria*,	hsa-let-7g-5p hsa-miR-107 hsa-miR-186-3p			[183]
		miRNA-30b miRNA-34a miRNA-146a miRNA-155			[184,185,186]
	*L. plantarum PS128*	miR-155-5p	SOCS1	inhibiting neuroinflammation response	[187]
Obesity	*B. eggerthi*	miR-130b-3p miR-185-5p miR-21-5p		correlating with BMI levels	[193]
	Tryptophan	miR-181		regulating energy expenditure and insulin sensitivity	[194]
	Ethanolamine	miR-101a-3p	Zo1	weakening intestinal barriers and induces gut permeability and inflammation	[195]
	SCFAs	miR-200c-3p miR-125a-5p miR-132-3p miR-223-3p			[196]
Cardiovascular disease		miR-17/92		promoting inflammation and atherosclerosis and CVD	[201]
	Protocatechuic acid	miRNA-10b		promoting reverse cholesterol transport and regresses atherosclerotic lesion	[202]
		microRNA-204	Sirtuin1	impairing endothelium-dependent vasorelaxation	[203]

## 6. Conclusions and Perspectives

In summary, studies have demonstrated the bidirectional interplay of gut microbiota and ncRNAs in regulating host homeostasis during development. Dysfunctions of the gut microbiota–ncRNA axis have been implicated in multiple diseases, including colorectal cancers, inflammatory bowel disease, neurological disease, obesity, and cardiovascular disease. Both the metabolites derived from gut microbiota and immunity participate in the regulation of the crosstalk between the gut microbiota–ncRNA axis and the host. However, the underlying molecular mechanisms are still fully understood. More detailed and direct evidence is needed. As ncRNAs are secreted into extracellular vesicles, fecal ncRNAs may act as valuable biomarkers for clinical diagnosis of gut-microbiota-related diseases. In the future, large cohorts of patients are needed to further prove this. As mentioned, the current studies mainly focus on miRNAs, but more attention should be paid to the investigation of the roles of lncRNAs and CircRNAs in crosstalk between microbiota and the host. Strategies targeting the gut microbiota–ncRNA axis, such as ncRNA mimics using nanovectors, ncRNA suppression via antisense oligonucleotides, and using small-molecule compounds to inhibit ncRNAs, as well as probiotics, prebiotics, and diets that modulate gut microbiota, may provide new clinical insights into treating human diseases.

## Figures and Tables

**Figure 1 genes-16-00208-f001:**
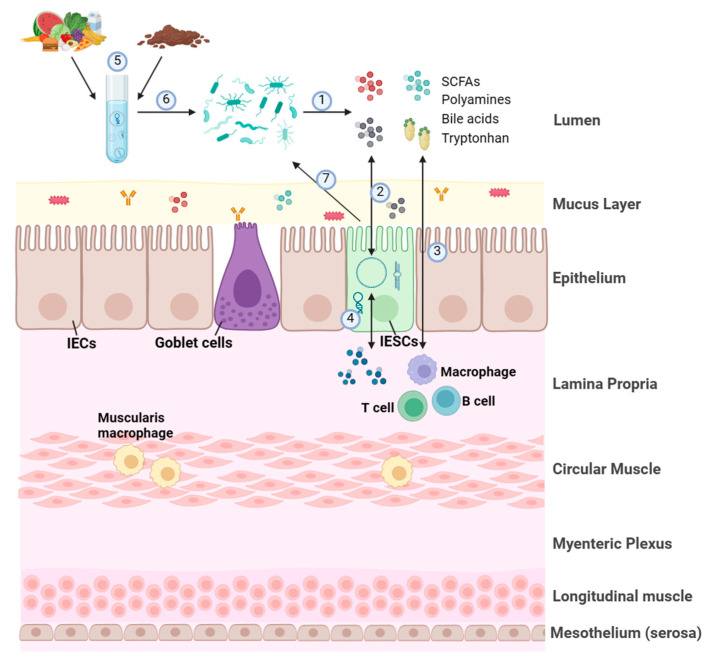
An overview of crosstalk among ncRNAs, gut microbiota, and host. The microbiota and its metabolites, including short-chain fatty acids (SCFSs), polyamines, bile acids, and tryptophan ①, regulate the expression of ncRNAs of intestinal epithelium cells (IECs) or intestinal epithelial stem cells (IESC) ②. Additionally, metabolites derived from microbiota also modulate gastrointestinal tract immunity system, such as T cells, B cells, and macrophages ③, in turn affecting the expression of ncRNAs of IECs ④. Accordingly, exosomal ncRNAs derived from feces and food ⑤ shape the composition and functions of gut microbiota ⑥. Moreover, changes in the expression of ncRNAs also affect the composition and functions of gut microbiota ⑦.

**Figure 2 genes-16-00208-f002:**
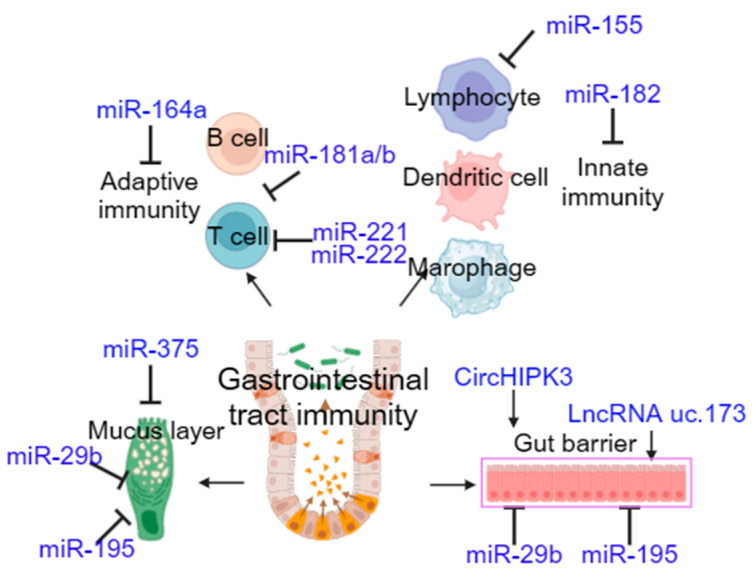
Summary of ncRNAs (in blue color) involved in regulating gastrointestinal tract immunity. Both adaptive (B cells and T cells) and innate immunity (lymphocytes, dendritic cells and macrophages), as well as mucosa and epithelial barrier, play important roles in gastrointestinal tract immunity, which is regulated by various ncRNAs.

**Figure 3 genes-16-00208-f003:**
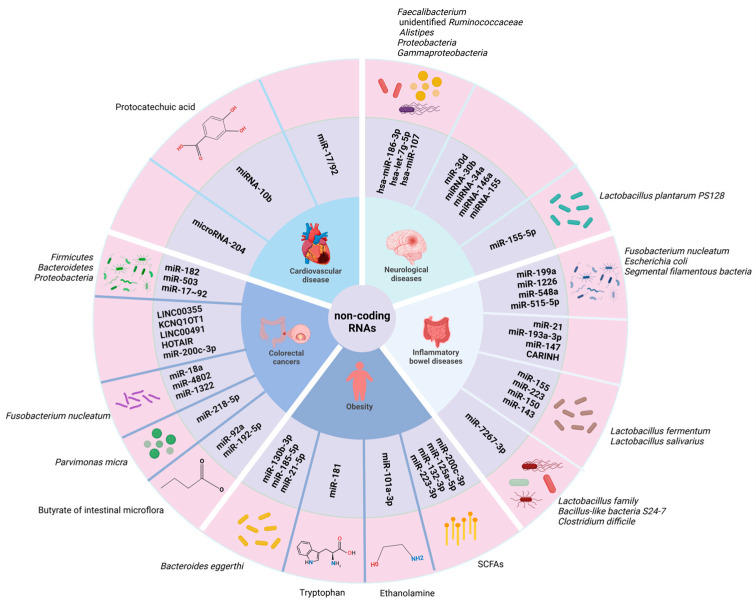
Summary of ncRNAs and related microbiota or derived metabolites involved in pathology of human diseases.

## Data Availability

No new data were created or analyzed in this study. Data sharing is not applicable to this article.

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
