# Peer review of "Bidirectional Interplay Among Non-Coding RNAs, the Microbiome, and the Host During Development and Diseases"

_genes, 2025, doi:10.3390/genes16020208_

Round 1
Reviewer 1 Report
Comments and Suggestions for Authors
Comments to authors:
The manuscript of a review article, which was written by Dr. Shanshan Nai et al, is interesting, revisiting relationships between ncRNAs and microbiota. Furthermore, authors discussed the possibility that cooperation of ncRNAs with gut bacteria induces biologically significant systems, including immunological responses. The topic itself is interesting because authors argued the possibilities of both ncRNAs and microbiota to be targets to treat various human diseases. However, authors had better revise the manuscript, adding a Table(s) that indicates specific associations between ncRNAs with specific bacteria.
Recommendation: Minor revision
General comments
This review article includes Figure 1. However, it needs to be edited providing essential information to understand the illustration. For example, the symbols for SCFSs, polyamines, bile acids, and Trp are not explained. Additionally, crosstalk between ncRNAs and bacteria is not shown by this Figure. Appropriate explanations are required. For example, what does that mean by drawings upon the test tube (upper left)? Put numbers on each arrow to sequentially explain the biologically essential events that are associated with ncRNA functions. Moreover, readers cannot understand at all by lines between Epithelium and ECG or Mast cells.
If this article is worthy to be not for reading but also to contribute to the progress in medical research or human health sciences, it will be a good idea to add some Tables or Figures in section 3. They will help to understand crosstalk between ncRNAs and specific bacteria. I would suggest that names of ncRNAs, biological functions, affected bacteria should be summarized with classification of the ncRNAs. In turn, the other Table would categorize bacteria that affect ncRNA metabolisms.
In section 5, authors are describing candidate ncRNAs that may be associated with human diseases. It would be a great help for readers comprehension if authors summarize or classify specific ncRNAs that may play a role in the pathogenesis of human diseases. I think both a Table and a Figure would be to help readers understand.
Minor comments
P1, L40: snRNAs should be explained.
P2, L46: NcRNAs; ncRNAs
P2, L66: miR-665 dysregulated; miR-665 to be dysregulated
P2, L78: Differential expressed; Differentially expressed
P2, L82, 97, P4, L151 and 158: E. coli or Escherichia coli? It should be unified all through the text.
P2, L87: PPARg; g should be typed in Greek
P2, L93: Liuet al; Liu et al. (add a comma)
P2, L96, P3, L105/107, P4, L151/158, and others: F. nucleatum and others; The names of organisms should be typed in italic.
P4, L139: host; What does it mean by “host”? In this article, most of the descriptions are for humans. However, if that were for other species, including mice, it should be explained.
P4, L158: Nissle 1917; Nissle et al.?
P4, L167: via should be typed in italic
P4, L169: [80-83]. Aberrant; delete an extra space
I would stop here to suggest such minor points. However, authors had better make best effort to find such minor points and correct.

Author Response
Reviewer 1#:
Comments to authors:
The manuscript of a review article, which was written by Dr. Shanshan Nai et al, is interesting, revisiting relationships between ncRNAs and microbiota. Furthermore, authors discussed the possibility that cooperation of ncRNAs with gut bacteria induces biologically significant systems, including immunological responses. The topic itself is interesting because authors argued the possibilities of both ncRNAs and microbiota to be targets to treat various human diseases. However, authors had better revise the manuscript, adding a Table(s) that indicates specific associations between ncRNAs with specific bacteria.
Response:
We would like to express our sincere thanks to the reviewer for the positive comments on our manuscript. We have added Tables and Figures that indicate specific associations between ncRNAs with specific bacteria according to reviewer’ suggestion.
Recommendation: Minor revision
General comments
This review article includes Figure 1. However, it needs to be edited providing essential information to understand the illustration. For example, the symbols for SCFSs, polyamines, bile acids, and Trp are not explained. Additionally, crosstalk between ncRNAs and bacteria is not shown by this Figure. Appropriate explanations are required. For example, what does that mean by drawings upon the test tube (upper left)? Put numbers on each arrow to sequentially explain the biologically essential events that are associated with ncRNA functions. Moreover, readers cannot understand at all by lines between Epithelium and ECG or Mast cells.
Response:
Thanks for the suggestion. We have deleted the nonessential lines between Epithelium and ECG or Mast cells, and revised the Figure to make it easier to understand. We have explained each symbol in the legend and put numbers on each arrow. The test tube means isolation of exosomes containing ncRNA from feces and food. We have added the description in the legend. Additionally, we also have shown the crosstalk between ncRNA and bacteria.
If this article is worthy to be not for reading but also to contribute to the progress in medical research or human health sciences, it will be a good idea to add some Tables or Figures in section 3. They will help to understand crosstalk between ncRNAs and specific bacteria. I would suggest that names of ncRNAs, biological functions, affected bacteria should be summarized with classification of the ncRNAs. In turn, the other Table would categorize bacteria that affect ncRNA metabolisms.
Response:
We thank the reviewer for the valuable comments. As suggested, we have added three Tables and one Figure in section 2 and 3. Table 1 summarizes the references about the status of microbiota, the corresponding differentially expressed host ncRNAs, their targets and functions. Table 2 summarizes the types of ncRNAs, the corresponding microbiota, and the biological functions. Table 3 summarizes the crucial metabolites derived from gut microbiota, and the corresponding host ncRNAs and their biological functions. Figure 2 reviews ncRNAs involved in regulating gastrointestinal tract immunity.
In section 5, authors are describing candidate ncRNAs that may be associated with human diseases. It would be a great help for readers comprehension if authors summarize or classify specific ncRNAs that may play a role in the pathogenesis of human diseases. I think both a Table and a Figure would be to help readers understand.
Response:
Thanks for the suggestion. We have added a Table (Table 4) and a Figure (Figure 3) to summarize ncRNAs invovled in the pathogenesis of human diseases in section 5.
Minor comments
P1, L40: snRNAs should be explained.
Response:
We apologized for the typos, and we have corrected “snRNAs” to “ncRNAs” in P1, L40.
P2, L46: NcRNAs; ncRNAs
Response:
We thank the reviewer for pointing out this issue. We have corrected NcRNAs to ncRNAs in P2 L46.
P2, L66: miR-665 dysregulated; miR-665 to be dysregulated
Response:
We have corrected “miR-665 dysregulated” to “miR-665 to be dysregulated” in P2, L66.
P2, L78: Differential expressed; Differentially expressed
Response:
We have corrected “Differential expressed” to “Differentially expressed” .
P2, L82, 97, P4, L151 and 158: E. coli or Escherichia coli? It should be unified all through the text.
Response:
We thank the reviewer for the correction, and we unify as Escherichia coli through the text in Line 83,104,163,176.
P2, L87: PPARg; g should be typed in Greek
Response:
We have revised PPARg to PPARγ.
P2, L93: Liuet al; Liu et al. (add a comma)
Response:
We have added a comma in Liu et al. In P2 L93.
P2, L96, P3, L105/107, P4, L151/158, and others: F. nucleatum and others; The names of organisms should be typed in italic.
Response:
We have written all names of organisms in italic in Line 103, 116, 243, 247, 308-309, 311-313.
P4, L139: host; What does it mean by “host”? In this article, most of the descriptions are for humans. However, if that were for other species, including mice, it should be explained.
Response:
We thank the reviewer for pointing out the issue, we have specify the host as human or murine in text.
P4, L158: Nissle 1917; Nissle et al.?
Response:
We have corrected the name of organisms as Escherichia coli Nissle 1917.
P4, L167: via should be typed in italic
Response:
We have changed “via” in italic.
P4, L169: [80-83]. Aberrant; delete an extra space
Response:
The extra space has been deleted.
I would stop here to suggest such minor points. However, authors had better make best effort to find such minor points and correct before publication.
Response:
We would like to express our sincere thank for the reviewer’s corrections, and we apologized again for the careless. We have checked through the whole manuscript to correct all the mistakes.

Reviewer 2 Report
Comments and Suggestions for Authors
This review examines the roles of non-coding RNAs and gut microbiome dysbiosis in development and disease, highlighting their interactions in conditions like colorectal cancer, inflammatory bowel diseases, and obesity. It suggests these interactions as potential biomarkers and therapeutic targets.
The manuscript is submitted at a time when there is a lot of research going on in this topic. There are many other reviews on the topic in the scientific literature, however this review of the literature may also be of interest to readers. However, the manuscript needs to be modified and improved in some aspects.
First of all, the manuscript focuses mainly on microRNAs for which more scientific studies have been published. I therefore suggest the authors to change the title to focus only on microRNAs specifically, and leave some comments on the remaining non-codingRNAs in the conclusions and perspectives section.
I also suggest to include some tables and images that help readers to better understand and remember the deregulated microRNAs described in the various sections of the manuscript.
There are a number of spelling inaccuracies and incorrect abbreviations in the manuscript, authors are asked to double-check the entire manuscript very carefully (scRNA, snRNA, interstinal epicellum cells, Mike, and so on....).

Author Response
Reviewer 2#:
This review examines the roles of non-coding RNAs and gut microbiome dysbiosis in development and disease, highlighting their interactions in conditions like colorectal cancer, inflammatory bowel diseases, and obesity. It suggests these interactions as potential biomarkers and therapeutic targets.
The manuscript is submitted at a time when there is a lot of research going on in this topic. There are many other reviews on the topic in the scientific literature, however this review of the literature may also be of interest to readers. However, the manuscript needs to be modified and improved in some aspects.
First of all, the manuscript focuses mainly on microRNAs for which more scientific studies have been published. I therefore suggest the authors to change the title to focus only on microRNAs specifically, and leave some comments on the remaining non-codingRNAs in the conclusions and perspectives section.
Response:
Firstly, we would like to thank the reviewer’s positive comments of our manuscript. As pointing out the issue by the reviewer, we indeed focus mainly on microRNAs for which more scientific studies have been published. However, we also review several recent high impact publications on lncRNAs and circRNAs, which are important in the filed. Therefore, we would like to keep the title of the manuscript. However, we have added some comments on the abstract, as well as conclusion and perspectives section to emphasize the emerging roles of lncRNAs and circRNAs, and the future research direction.
I also suggest to include some tables and images that help readers to better understand and remember the deregulated microRNAs described in the various sections of the manuscript.
Response:
We thank the reviewer for the valuable comments. As suggested, we have added several Tables and Figures in section 2, 3 and 5. Table 1 summarizes the references about the status of microbiota, the corresponding differentially expressed host ncRNAs, their targets and functions. Table 2 summarizes the types of ncRNAs, the corresponding microbiota, and the biological functions. Table 3 summarizes the crucial metabolites derived from gut microbiota, and the corresponding host ncRNAs and their biological functions. Figure 2 reviews ncRNAs involved in regulating gastrointestinal tract immunity. Additionally, we also have added a Table (Table 4) and a Figure (Figure 3) to summarize ncRNAs invovled in the pathogenesis of human diseases in section 5.
There are a number of spelling inaccuracies and incorrect abbreviations in the manuscript, authors are asked to double-check the entire manuscript very carefully (scRNA, snRNA, interstinal epicellum cells, Mike, and so on....).
Response:
We apologized for the careless. We have checked carefully through the whole manuscript to correct all the mistakes.

Reviewer 3 Report
Comments and Suggestions for Authors
In the manuscript entitled “Bidirectional interplay among noncoding RNA-microbiome-host during development and diseases” authors review the current molecular understanding and the possible effects on host pathology of the crosstalk between noncoding RNAs and the gut microbiome.
General comment:
Authors review an interesting topic. Nonetheless, according to Gregory et al. 2018 (doi:10.1016/j.hlc.2018.03.027) the main task of a review “is to not only summarize the relevant literature but to also analyze it, to provide a critical discussion of it, and to identify methodological problems”. The review presents relevant information on a vast and complex topic. However, original contribution and critical discussion of the current manuscript to the subject area compared with other recent published reviews on the same topic seems to be limited.
Specific comments:
1. Line 40: “scRNA”. Check abbreviation.
2. Fig 1. All abbreviations in figure 1 should be explained in the legend. The significance of the lines should be provided. Clearly identify the meaning of the test tube.
3. Line 70: “miRNAs respond to microbiota in a highly cell type-specific manner”. Please make clear how miRNAs can respond to microbiota.
4. Line 102: “dietary-derived exosomal ncRNAs exert gut microbiota modulation”. How microbiota signature/composition can impinge on diet ncRNAs uptake efficiency?
5. Line 112: “Mike exosome-derived miRNA”. Revise.
6. Line 147: “Vitamin E alleviates cardiac hypertrophy and fibrosis in mice likely through down-regulation of miR-21 and miR-499 and upregulation of miR-210”. Different sources of vitamin E (including diet) should be considered.
7. Line 176: “Several studies have demonstrated ncRNA involved in gastrointestinal tract immunity”. In addition to gut microbiome also host genetic factors and environmental factors play a key role in ncRNA modulation in both homeostatic and pathological conditions.
8. Line 233 “Extensive studies have illustrated crucial functions of ncRNA in gene expression control during both developmental and differentiation processes”. Epigenetic mechanisms, including chromatin accessibility and DNA methylation, should also be briefly acknowledged.
9. Line 14: “potential roles of noncoding RNAs and gut microbiome interactions as biomarkers and therapeutic targets for clinical diagnosis and treatment.” Authors may consider to expand these interesting aspects anticipated in the abstract.

Author Response
Reviewer 3#:
In the manuscript entitled “Bidirectional interplay among noncoding RNA-microbiome-host during development and diseases” authors review the current molecular understanding and the possible effects on host pathology of the crosstalk between noncoding RNAs and the gut microbiome.
General comment:
Authors review an interesting topic. Nonetheless, according to Gregory et al. 2018 (doi:10.1016/j.hlc.2018.03.027) the main task of a review “is to not only summarize the relevant literature but to also analyze it, to provide a critical discussion of it, and to identify methodological problems”. The review presents relevant information on a vast and complex topic. However, original contribution and critical discussion of the current manuscript to the subject area compared with other recent published reviews on the same topic seems to be limited.
Response:
We would like to express our sincere thank for pointing out the critical issue by the reviewer. In the revised manuscript, we have added some critical discussion about the current studies and perspectives for future research.
Specific comments:
- Line 40: “scRNA”. Check abbreviation.
Response:
We have corrected “scRNA” into “ncRNA”, as suggested. In addition, we also have carefully checked the whole manuscript to correct spelling and abbreviations errors.
- Fig 1. All abbreviations in figure 1 should be explained in the legend. The significance of the lines should be provided. Clearly identify the meaning of the test tube.
Response:
We thank the reviewer for the valuable suggestions. We have revised the figure to make it more clearly understand. We have explained all the abbreviations and test tube in the legend, as well as numbered each line and explained them in the legend.
- Line 70: “miRNAs respond to microbiota in a highly cell type-specific manner”. Please make clear how miRNAs can respond to microbiota.
Response:
We apologized for unclearity. In the reference, the author investigated differentially expressed miRNAs in different types of IECs compared GF and conventional states, and found the miRNAs in IESCs exhibited more distinctly expressed. We have changed the statement to make it clear.
- Line 102: “dietary-derived exosomal ncRNAs exert gut microbiota modulation”. How microbiota signature/composition can impinge on diet ncRNAs uptake efficiency?
Response:
We have changed the statement to “dietary-derived exosomal ncRNAs can modulate gut microbiota signature or composition, thereby affecting human or murine physiology” to make it clearer.
- Line 112: “Mike exosome-derived miRNA”. Revise.
Response:
We thank the reviewer for pointing out this issue. The mistakes have been corrected.
- Line 147: “Vitamin E alleviates cardiac hypertrophy and fibrosis in mice likely through down-regulation of miR-21 and miR-499 and upregulation of miR-210”. Different sources of vitamin E (including diet) should be considered.
Response:
Thanks for the suggestion.We have revised this to “Supplemented with recombinant Vitamin E in mice reduced the levels of miR-21 and miR-499 and increased the expression of miR-210, thereby alleviating cardiac hyper-trophy and fibrosis”.
- Line 176: “Several studies have demonstrated ncRNA involved in gastrointestinal tract immunity”. In addition to gut microbiome also host genetic factors and environmental factors play a key role in ncRNA modulation in both homeostatic and pathological conditions.
Response:
We agree with the reviewer’ comments. We have added the sentence “In addition to host genetic factors and environmental factors, studies have shown that changes of gut microbiome status can affect the host ncRNA expression profiling” in Page 2 Line 66.
- Line 233 “Extensive studies have illustrated crucial functions of ncRNA in gene expression control during both developmental and differentiation processes”. Epigenetic mechanisms, including chromatin accessibility and DNA methylation, should also be briefly acknowledged.
Response:
Thanks the suggestion from the reviewer. We have added these comments in Page 7 Line 272, “Extensive studies have illustrated crucial functions of ncRNAs, as well as other epigenetic machanisms including chromatin accessibility, DNA methylation and his-tone modification during both developmental and differentiation processes, such as stem cell maintenance, embryogenesis, organogenesis, dosage compensation and genomic imprinting”.
9.Line 14: “potential roles of noncoding RNAs and gut microbiome interactions as biomarkers and therapeutic targets for clinical diagnosis and treatment.” Authors may consider to expand these interesting aspects anticipated in the abstract.
Response:
As suggested, we have expand these aspects in the abstract.

Round 2
Reviewer 3 Report
Comments and Suggestions for Authors
Authors addressed my comments and did their best to address them, incorporating into the revised manuscript their response.